# Stark Widths of Yb III and Lu IV Spectral Lines

**Milan S. Dimitrijević** [1,2,†] 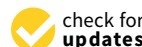

1    Astronomical Observatory, Volgina 7, 11060 Belgrade, Serbia; mdimitrijevic@aob.rs; Tel.: +381-64-297-8021
2    Sorbonne Université, Observatoire de Paris, Université PSL, CNRS, LERMA, F-92190 Meudon, France
†    Current address: Astronomical Observatory, Volgina 7, 11060 Belgrade, Serbia.

**Abstract:** Rare Earth Elements are important for stellar atmosphere analysis but the corresponding Stark broadening data are scarce. For Yb III and Lu IV theoretical as well as experimental data on Stark broadening parameters of spectral lines are absent in the literature. Using the modified semiempirical method of Dimitrijević and Konjević, we determined Stark widths for four Yb III and four Lu IV transitions, belonging to the erbium isoelectronic sequence. The obtained results are also used to discuss similarities between homologous transitions in the erbium isoelectronic sequence. We note as well that calculated widths will be implemented in the STARK-B database which is also a part of the Virtual Atomic and Molecular Data Center.

**Keywords:** Stark broadening; line profiles; atomic data

## 1. Introduction

The development and use of space observations with satellite-born instruments increased the importance of trace elements for investigation of stellar spectra. As an example, Rauch et al. [1] highlighted the need of the reliable Stark broadening data for as many spectral lines as possible of different atoms and ions, since they "are of crucial importance for sophisticated analysis of stellar spectra by means of NLTE model atmospheres." Such data are not only useful for analysis, synthesis and research of high resolution spectra obtained from space born instruments, they are very useful for laboratory plasma diagnostics as well as for investigation of various plasmas in laser physics, fusion research and for plasma-based technologies.

Triply charged lutetium ion (Lu IV ) as well as doubly charged ytterbium ion (Yb III) belong to the Er isoelectronic sequence and for both the electronic configuration of ground state is $[Kr]4d^{10}5s^25p^64f^{14}$. They both belong to the Rare Earth Elements (REE) and are part of the REE peak in the stellar abundance distribution of chemical elements. So they are of astrophysical importance and both were found in stellar spectra. For example Hawkins et al. [2] found the Yb II line in the Arcturus spectrum and Afsar et al. [3] in the spectra of HIP 54048, HIP114809 and HIP57748 stars. Lu II spectral lines were observed, e.g., in the spectrum of CS3108-001 star [4] and stellar abundances for both ytterbium and lutetium have been determined in Roederer et al. [5]. Spectral lines of ions of both elements are found also in the spectrum of Przybylski's star [6] and one can expect that YbIII and Lu IV lines, will be observed in the future. Moreover, they are of interest as well for theoretical consideration and modelling of stellar plasma (atmospheres and subphotospheric layers) where the electron density is sufficiently high (in present case higher than $10^{15}$–$10^{16}$ cm$^{-3}$) and for the corresponding radiative transfer calculations. Stark broadening data for the considered ions may be of interest in laser physics and laser produced plasma since both are used in laser technology. Of course such data are of interest as well in laboratory plasma investigations and diagnostics.

Recently Stark broadening data for 27 Lu III spectral lines, calculated by using the modified semiempirical method, have been published [7]. Since for Yb III and Lu IV neither theoretical nor experimental data for Stark broadening exist, in this work are presented results of our calculations of full widths at half intensity maximum (FWHM), due to impacts with electrons, for spectral lines of these ions, using the modified semiempirical method (MSE) [8–10].

## 2. The Modified Semiempirical Method

The electron impact full width (FHWM) of an isolated ion line within the modified semiempirical (MSE) approach [8], may be expressed in the form:

$$w_{MSE} = N \frac{4\pi}{3c} \frac{\hbar^2}{m^2} \left(\frac{2m}{\pi kT}\right)^{1/2} \frac{\lambda^2}{\sqrt{3}} \cdot \{\sum_{\ell_i \pm 1} \sum_{L_{i'} J_{i'}} \vec{\Re}^2_{\ell_i,\ell_i \pm 1} \widetilde{g}(x_{\ell_i,\ell_i \pm 1}) +$$

$$\sum_{\ell_f \pm 1} \sum_{L_{f'} J_{f'}} \vec{\Re}^2_{\ell_f,\ell_f \pm 1} \widetilde{g}(x_{\ell_f,\ell_f \pm 1}) + (\sum_{i'} \vec{\Re}^2_{ii'})_{\Delta n \neq 0} g(x_{n_i,n_i+1}) + (\sum_{f'} \vec{\Re}^2_{ff'})_{\Delta n \neq 0} g(x_{n_f,n_f+1})\}. \tag{1}$$

where $i$ denotes the initial level and $f$ the final one. In the case of $J_1 j$ coupling, used to describe Yb III and Lu IV terms, the square of the matrix element $\{\vec{\Re}^2[n_k \ell_k j_k J_k, (\ell_k \pm 1) j_{k'} J_{k'}], \quad k = i, f\}$ may be presented as

$$\vec{\Re}^2[n_k \ell_k j_k J_k, n_k(\ell_k \pm 1) j_{k'} J_{k'}] = \frac{\ell_>}{2J_k + 1} Q[\ell_k j_k, (\ell_k \pm 1) j_{k'}] Q(J_k, J_{k'}) [R^{n_k^*(\ell_k \pm 1)}_{n_k^* \ell_k}]^2. \tag{2}$$

Here, $\ell_> = \max(\ell_k, \ell_k \pm 1)$ and

$$\left(\sum_{k'} \vec{\Re}^2_{kk'}\right)_{\Delta n \neq 0} = \left(\frac{3n_k^*}{2Z}\right)^2 \frac{1}{9} (n_k^{*2} + 3\ell_k^2 + 3\ell_k + 11) \tag{3}$$

In Equation (1)

$$x_{\ell_k,\ell_{k'}} = \frac{E}{\Delta E_{\ell_k,\ell_{k'}}}, \quad k = i, f$$

$E = \frac{3}{2}kT$ is the electron kinetic energy and $\Delta E_{\ell_k,\ell_{k'}} = |E_{\ell_k} - E_{\ell_{k'}}|$ is the energy difference between levels $\ell_k$ and $\ell_k \pm 1$ ($k = i, f$),

$$x_{n_k,n_k+1} \approx \frac{E}{\Delta E_{n_k,n_k+1}},$$

where for $\Delta n \neq 0$, the energy difference between energy levels with $n_k$ and $n_k + 1$, $\Delta E_{n_k,n_k+1}$ is approximated as

$$\Delta E_{n_k,n_k+1} = 2Z^2 E_H / n_k^{*3}, \tag{4}$$

$n_k^* = [E_H Z^2 / (E_{ion} - E_k)]^{1/2}$ is the effective principal quantum number, $Z$ is the residual ionic charge (e.g., $Z = 1$ for neutrals) and $E_{ion}$ is the appropriate spectral series limit. $N$ and $T$ are electron density and temperature, and $Q(\ell j, \ell' j')$, $Q(J, J')$ multiplet and line factors. With $g(x)$ [11,12] and $\widetilde{g}(x)$ [8] are denoted the Gaunt factors. The calculation of radial integrals $[R^{n_k^* \ell_k \pm 1}_{n_k^* \ell_k}]$ have been performed within the Coulomb approximation in accordance with Bates and Damgaard [13] and using the tables of Oertel and Shomo [14]. When the corresponding data are absent in Oertel and Shomo [14], the needed radial integrals can be calculated according to Ref. Van Regemorter et al. [15].

## 3. Results and Discussion

The atomic energy levels of Yb III and Lu IV were from Martin et al. [16], Kramida et al. [17], while the matrix elements were calculated within the Coulomb approximation [13]. Calculation of FWHM due to electrons as perturbers (Stark width), has been performed by using the modified semiempirical

method [8] (see also e.g., [10]). We assumed that the energy levels are pure so that the configuration mixing is neglected. Consequently the transitions where this effect can be neglected were chosen.

The obtained results for four Yb III and four Lu IV transitions are presented in Table 1 for perturber density of $10^{17}$ cm$^{-3}$ and temperatures from 5000 K up to 160,000 K. We choose this temperature range due to its interest for applications in astrophysics, laboratory plasma, for lasers and laser produced plasma. In the case of perturber densities lower than $10^{17}$ cm$^{-3}$ the extrapolation is linear. For higher perturber densities one can also use a linear extrapolation checking that the influence of Debye screening may be neglected or it is reasonably small. The wavelengths in Table 1 are calculated from the corresponding energy levels given as the input so that they may differ from the observed ones. Additionally is provided the ratio of $E = 3kT/2$, and the energy difference of initial or final and the closest perturbing level, $\Delta E$. It is calculated for $T$ = 10,000 K:

$$\Delta E = \text{Max}[E/\Delta E_{i,i'}, E/\Delta E_{f,f'}, E/\Delta E_{n_i,n_i+1}, E/\Delta E_{n_f,n_f+1}] \tag{5}$$

For a given temperature the value of $3kT/2\Delta E = 1$ represents the threshold for the relevant inelastic transition. For values lower than one, elastic collisions dominate. For values larger than approximately 50, the high temperature limit approximation can be applied.

We did not find any experimental or theoretical data for Yb III and Lu IV for comparison with our results.

For an analysis of similarities and regular behavior of Stark widths in the case of $J_1j$ coupling, we need to convert obtained line widths from Å units to angular frequency units, in order to exclude the influence of wavelength. This can be done using the expression:

$$W(\text{Å}) = \frac{\lambda^2}{2\pi c} W(s^{-1}) \tag{6}$$

here $c$ is the speed of light.

The corresponding values are given in Table 1 as well. We can see that the Stark widts for $(J_1, 3/2)$ transitions are higher from the values for $(J_1, 1/2)$ for 6% in the case of Yb III and for 11% in the case of Lu IV. Surprisingly, for both ions Stark widths for $(5/2,j)$ and $(7/2,j)$ are practically identical. This can be used if we have the Stark width value for one of such lines and not for the other which we need.

It is stated in Majlinger et al. [7] that the theoretical resolving power of the high-resolution echelle spectrometer for the Keck Ten—Meter Telescope is of the order of >250,000, but that practical realizations may be approximately 36,000. Resolving power for Stark widths from Table 1 should be for Yb III from 12,056 at T = 5000 K to 68,825 at T = 160,000 K and for Lu IV from 28,356 at T = 5000 K to 165,811 for T = 160,000 K. We can see that exist condition when we can observe the influence of Stark broadening on Yb III and Lu IV specral lines with large terrestrial telescopes.

The results for Stark widths of Yb III and Lu IV spectral lines obtained in this work and presented in Table 1, will be included in the STARK-B database [18,19], which has a principal aim to serve for the investigations of the plasma of stellar atmospheres and for stellar spectra analysis. Of course these data could be very useful for diagnostics of laboratory plasmas, as and for various research on laser produced inertial fusion plasma and plasmas in different technologies.

**Table 1.** FWHM—Full Width at Half intesity Maximum *W* in Å and in s$^{-1}$ for Yb III and Lu IV spectral lines, for a perturber density of $10^{17}$ cm$^{-3}$ and temperatures from 5000 to 160,000 K. Calculated wavelength ($\lambda$) of the transitions (in Å) is also given.

| Transition | *T* [K] | *W* [Å] | *W* [$10^{12}$ s$^{-1}$] |
|---|---|---|---|
| YbIII $4f^{13}(^2F^o_{7/2})6s_{1/2}(7/2,1/2)^o - 4f^{13}(^2F^o_{7/2})6p_{1/2}(7/2,1/2)$ | 5000 | 0.221 | 0.586 |
| | 10,000 | 0.156 | 0.414 |
| $\lambda$ = 2664.3 Å | 20,000 | 0.110 | 0.292 |
| 3kT/2ΔE = 0.278 | 40,000 | 0.0780 | 0.207 |
| | 80,000 | 0.0556 | 0.148 |
| | 160,000 | 0.0446 | 0.118 |
| YbIII $4f^{13}(^2F^o_{7/2})6s_{1/2}(7/2,1/2)^o - 4f^{13}(^2F^o_{7/2})6p_{3/2}(7/2,3/2)$ | 5000 | 0.166 | 0.600 |
| | 10,000 | 0.118 | 0.424 |
| $\lambda$ = 2285.0 Å | 20,000 | 0.0831 | 0.300 |
| 3kT/2ΔE = 0.278 | 40,000 | 0.0588 | 0.212 |
| | 80,000 | 0.0416 | 0.150 |
| | 160,000 | 0.0332 | 0.120 |
| YbIII $4f^{13}(^2F^o_{5/2})6s_{1/2}(5/2,1/2)^o - 4f^{13}(^2F^o_{5/2})6p_{1/2}(5/2,1/2)$ | 5000 | 0.221 | 0.591 |
| | 10,000 | 0.157 | 0.418 |
| $\lambda$ = 2657.0 Å | 20,000 | 0.111 | 0.296 |
| 3kT/2ΔE = 0.277 | 40,000 | 0.0783 | 0.209 |
| | 80,000 | 0.0558 | 0.149 |
| | 160,000 | 0.0447 | 0.119 |
| YbIII $4f^{13}(^2F^o_{5/2})6s_{1/2}(5/2,1/2)^o - 4f^{13}(^2F^o_{5/2})6p_{3/2}(5/2,3/2)$ | 5000 | 0.169 | 0.607 |
| | 10,000 | 0.119 | 0.430 |
| $\lambda$ = 2288.9 Å | 20,000 | 0.0845 | 0.304 |
| 3kT/2ΔE = 0.277 | 40,000 | 0.0597 | 0.215 |
| | 80,000 | 0.0423 | 0.152 |
| | 160,000 | 0.0338 | 0.121 |
| LuIV $4f^{13}(^2F^o_{7/2})6s_{1/2}(7/2,1/2)^o - 4f^{13}(^2F^o_{7/2})6p_{1/2}(7/2,1/2)$ | 5000 | 0.0742 | 0.316 |
| | 10,000 | 0.0525 | 0.223 |
| $\lambda$ = 2104.4 Å | 20,000 | 0.0371 | 0.158 |
| 3kT/2ΔE = 0.219 | 40,000 | 0.0262 | 0.112 |
| | 80,000 | 0.0186 | 0.0789 |
| | 160,000 | 0.0140 | 0.0594 |
| LuIV $4f^{13}(^2F^o_{7/2})6s_{1/2}(7/2,1/2)^o - 4f^{13}(^2F^o_{7/2})6p_{3/2}(7/2,3/2)$ | 5000 | 0.0574 | 0.350 |
| | 10,000 | 0.0406 | 0.248 |
| $\lambda$ = 1757.6 Å | 20,000 | 0.0287 | 0.175 |
| 3kT/2ΔE = 0.219 | 40,000 | 0.0203 | 0.124 |
| | 80,000 | 0.0144 | 0.0876 |
| | 160,000 | 0.0106 | 0.0649 |
| LuIV $4f^{13}(^2F^o_{5/2})6s_{1/2}(5/2,1/2)^o - 4f^{13}(^2F^o_{5/2})6p_{1/2}(5/2,1/2)$ | 5000 | 0.0735 | 0.316 |
| | 10,000 | 0.0520 | 0.223 |
| $\lambda$ = 2093.0 Å | 20,000 | 0.0368 | 0.158 |
| 3kT/2ΔE = 0.218 | 40,000 | 0.0260 | 0.112 |
| | 80,000 | 0.0184 | 0.0790 |
| | 160,000 | 0.0138 | 0.0594 |
| LuIV $4f^{13}(^2F^o_{5/2})6s_{1/2}(5/2,1/2)^o - 4f^{13}(^2F^o_{5/2})6p_{3/2}(5/2,3/2)$ | 5000 | 0.0573 | 0.350 |
| | 10,000 | 0.0405 | 0.247 |
| $\lambda$ = 1757.2 Å | 20,000 | 0.0287 | 0.175 |
| 3kT/2ΔE = 0.218 | 40,000 | 0.0203 | 0.124 |
| | 80,000 | 0.0143 | 0.0874 |
| | 160,000 | 0.0106 | 0.0648 |

We wish to underline that STARK-B database is one of 33 databases with atomic and molecular data which enter in the Virtual Atomic and Molecular Data Center (VAMDC) [20,21], in order to provide an e-platform for more effective search and mining of atomic and molecular data.

**Funding:** This research received no external funding.

**Conflicts of Interest:** The author declares no conflict of interest.

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
