# Peer review of "Stark Widths of Yb III and Lu IV Spectral Lines"

_atoms, doi:10.3390/atoms7010010_

Reviewer 1 Report

In the paper STARK WIDTHS OF Yb III AND Lu IV SPECTRAL LINES written by Milan S. Dimitrijević , the author presents the calculation of full widths at half intensity maximum (FWHM) of Rare Earth Elements Yb III and Lu IV ions spectral lines broadened by the electron impact (Stark broadening).  Stark widths are important in plasma diagnostic (electron densities and temperature) as well as for stellar atmosphere modeling.

Author of this paper is an expert in the field of Stark broadening. Dimitrijević and Konjević (1980) simplified Griem procedure for Stark broadening parameters determination. They carefully chose Gaunt factors for elastic and inelastic collisions in order to get a good agreement between theoretical and existing experimental values of Stark widths. Using atomic energy levels from the literature as the input data this numerically efficient method produces correct results. Dimitrijević and Popović (2001) presented a Modified semi empirical method (MSM) and pointed out that the oscillatory strength of the relevant atomic transitions (if they exist in the literature) can be used as empirical input data in the calculation. In the section 2 of this paper the main points of MSM procedure are briefly explained.

Calculated Stark widths (FWHM) of four Yb III and Lu IV spectral lines are presented in the section 3. The calculation was performed for the perturber density 1017 cm-3 in the electronic temperature range 5000K to 160000K. Because Yb III and Lu IV ions are members of erbium isoelectronic series there are similarities between their atomic transitions what was discussed in section 3.  For the Stark widths presented in this paper there are no other theoretical or experimental data in the literature, and therefore data from this paper are a valuable contribution to Stark databases.

The article is well organized and written in a clear way. I find only two typos that should be corrected; line 2: scarse should be scarce and line 23: stella spectra should be stellar spectra.

The results of this paper are useful for plasma diagnostic and stellar atmosphere modeling and I recommend this paper for publication. In my opinion the paper is complete and should be published as it is.

Author Response

I am very grateful to the reviewer's opinion on the manuscript. The found typos are corrected. Thank you very much

Reviewer 2 Report

Review of manuscript:

STARK WIDTHS OF Yb III AND Lu IV SPECTRAL LINES
by Milan S. Dimitrijević

It is still within minor grammatical typos and results are publishable as they stand,
except for minor augmentations regarding Table 1 and possible typo in Equation (5).
Please correct for some minor omissions, such as missing letters in ‘innovation’ and ‘of course’. A sentence in introduction on what is ‘dipper atmosphere’ in terms of Thomson optical depth of ~0.1 will help non-astrophysics readers.

line#, page#, Section: original text, suggested text.
Line 1, page 1, Abstract: corresponsing---corresponding
Line 2, page 1, Abstract: scarse---scarce
Line 17, page 1, Introduction: useful and for---useful for
Line 18, page 1, Introduction: phyics---physics
Line 18, page 1, Introduction: technologies based on plasma---plasma-based technologies.
Line 20, page 1, Introduction: 4d^10 5s^2 5p^6 4f^14--- [Kr]4d^10 5s^2 5p^6 4f^14
Line 23, page 1, Introduction: stella---stellar
Line 30, page 1, Introduction: and for laser physics---in laser physics
Line 32, page 1, Introduction: and for laboratory---in laboratory
Line 56, page 2, The Modified...: there is no the corresponding---there are no corresponding
Line 63, page 3, 3. Results and ...: inovations---innovations
Line 71, page 3, 3. Results and ...: Additinally---Additionally

Equation (5) on page 3: The energy difference is chosen as a maximum of four terms, first teo of which are ratios of energies and other two are energies.
Is there a typo, like perhaps instead of ratio '/', the difference '-' symbol is to be used?

Line 96, page 3, 3. Results and ...: Of cours---Of course

Table 1 on page 4: It will help readers if the reported state labels also indicate the largest two mixing coefficient with which is such label obtained.
For example, is the YbIII 4f^13 (^2F^o_7/2)6s_1/2(7/2,1/2)^o a 100% in chosen J1j-coupling scheme, or is it perhaps 80% with 20% contribution of some other
near-coupled state? This will help readers gauge about the completeness and the size of the configuration basis necessary to reproduce the reported widths.

Author Response

The author would like to express his appreciation to the referee for the kind and valuable comments on our manuscript. We have revised our manuscript according to the referee's comments.

Reviewer:
A sentence in introduction on what is ‘dipper atmosphere’ in terms of Thomson optical depth of ~0.1 will help non-astrophysics readers.

Answer:
"theoretical consideration and modelling of dipper layers of stellar atmospheres"
is replaced by:
"theoretical consideration and modelling of stellar plasma (atmospheres and subphotospheric layers) where the electron density is sufficiently high (in present case higher than 10$^{15}$ - 10$^{16}$ cm$^-3}$)"

Reviewer:
line#, page#, Section: original text, suggested text.
Line 1, page 1, Abstract: corresponsing---corresponding
Line 2, page 1, Abstract: scarse---scarce
Line 17, page 1, Introduction: useful and for---useful for
Line 18, page 1, Introduction: phyics---physics
Line 18, page 1, Introduction: technologies based on plasma---plasma-based technologies.
Line 20, page 1, Introduction: 4d^10 5s^2 5p^6 4f^14--- [Kr]4d^10 5s^2 5p^6 4f^14
Line 23, page 1, Introduction: stella---stellar
Line 30, page 1, Introduction: and for laser physics---in laser physics
Line 32, page 1, Introduction: and for laboratory---in laboratory
Line 56, page 2, The Modified...: there is no the corresponding---there are no corresponding
Line 63, page 3, 3. Results and ...: inovations---innovations
Line 71, page 3, 3. Results and ...: Additinally---Additionally

Author:
Corrected. Thank you.

Reviewer:
Equation (5) on page 3: The energy difference is chosen as a maximum of four terms, first teo of which are ratios of energies and other two are energies.
Is there a typo, like perhaps instead of ratio '/', the difference '-' symbol is to be used?

Author:
Thank you very much indeed for careful reading. The equation is corrected.

Reviewer:
Line 96, page 3, 3. Results and ...: Of cours---Of course

Author:
Corrected. Thank you.

Reviewer:
Table 1 on page 4: It will help readers if the reported state labels also indicate the largest two mixing coefficient with which is such label obtained.
For example, is the YbIII 4f^13 (^2F^o_7/2)6s_1/2(7/2,1/2)^o a 100% in chosen J1j-coupling scheme, or is it perhaps 80% with 20% contribution of some other
near-coupled state? This will help readers gauge about the completeness and the size of the configuration basis necessary to reproduce the reported widths.

Author:
In the first paragraph of "Results and discussion is added:
We assumed that the energy levels are pure so that the configuration mixing is neglected.
Consequently the transitions where this effect can be neglected were chosen.